# Simultaneous Intake of Chlorella and Ascidian Ethanolamine Plasmalogen Accelerates Activation of BDNF–TrkB–CREB Signaling in Rats

**DOI:** 10.3390/molecules29020357

**Published:** 2024-01-11

**Authors:** Hideo Takekoshi, Masaki Fujishima, Taiki Miyazawa, Ohki Higuchi, Takahiko Fujikawa, Teruo Miyazawa

**Affiliations:** 1Production and Development Department, Sun Chlorella Corp., Kyoto 600-8177, Japan; mfujishima@sunchlorella.co.jp; 2Food Biotechnology Platform Promoting Project, New Industry Creation Hatchery Center (NICHe), Tohoku University, Sendai 980-8579, Japan; taiki.miyazawa.b3@tohoku.ac.jp (T.M.); higuchi@hokkaido-bpi.co.jp (O.H.); teruo.miyazawa.a7@tohoku.ac.jp (T.M.); 3Biodynamic Plant Institute Co., Ltd., Sapporo 004-0015, Japan; 4Laboratory of Molecular Prophylaxis and Pharmacology, Graduate School of Pharmaceutical Sciences, Suzuka University of Medical Science, Suzuka 513-8670, Japan; t-fuji@suzuka-u.ac.jp

**Keywords:** Alzheimer’s disease, antioxidant, ascidian, brain-derived neurotrophic factor, Chlorella, docosahexaenoic acid, glutamate receptors, hippocampus, lutein, plasmalogen

## Abstract

Brain-derived neurotrophic factor (BDNF) plays an important role in neurogenesis, synaptic plasticity, and cognition. BDNF is a neurotrophin that binds to tropomyosin receptor kinase B (TrkB), a specific receptor on target cell surfaces; it acts on neuronal formation, development, growth, and repair via transcription factors, such as cAMP response element-binding protein (CREB), and it is involved in learning and memory. BDNF expression is decreased in patients with Alzheimer’s disease (AD). Exercise and the intake of several different foods or ingredients can increase BDNF expression, as confirmed with lutein, xanthophylls (polar carotenoids), and ethanolamine plasmalogen (PlsEtn), which are present at high levels in the brain. This study examined the effects of combining lutein and PlsEtn using lutein-rich Chlorella and ascidian extracts containing high levels of PlsEtn bearing docosahexaenoic acid, which is abundant in the human brain, on the activation of the BDNF–TrkB–CREB signaling pathway in the hippocampus of Sprague-Dawley rats. Although activation of the BDNF–TrkB–CREB signaling pathway in the hippocampus was not observed in Chlorella or ascidian PlsEtn monotherapy, activation was observed with combination therapy at an equal dose. The results of this study suggest that the combination of Chlorella and ascidian PlsEtn may have a preventive effect against dementia, including AD.

## 1. Introduction

Dementia is an increasing social problem, not only in Japan but also worldwide. The number of patients with dementia worldwide as of 2019 was approximately 57 million [1]. In Japan, the number of patients aged 65 years or older with dementia reached approximately 6 million by 2020 and is expected to increase to approximately 8 million by 2040 [2]. Efforts to prevent the development of dementia are gaining increased attention, including the development of drugs and preventive treatments using foods and ingredients that can be consumed daily. There is also a prediction that in the post-COVID-19 era, food components that maintain the human body will become more important as the demand of the food industry increases [3].

Among patients with dementia in Japan, 65–70% have Alzheimer’s disease (AD) [4]. The leading theory on the cause of AD is the amyloid hypothesis [5], in which amyloid-β, a component protein of amyloid plaques, accumulates in the brain and causes neuronal degeneration; however, this hypothesis has not been fully elucidated. Plasmalogen concentrations decrease with atrophy in the brains of patients with AD [6,7]. The brain is rich in lipids in comparison with other organs. The human brain contains approximately 65% lipids by dry weight, and its lipid composition is highly specific; for example, plasmalogen, which accounts for only approximately 5% of phosphatidylethanolamine in the liver, accounts for 50–65% of phosphatidylethanolamine in the brain [6,7,8]. Plasmalogen is involved in membrane fusion during synaptic transmission in the nervous system due to its propensity for hexagonal phase formation [9].

*Chlorella pyrenoidosa* (Chlorella), a unicellular green alga, is used as a dietary supplement and functional food in many countries because it contains abundant proteins, vitamins, and minerals [10,11]. In addition, compared to other foods, Chlorella has a very high lutein content, a major antioxidant in human erythrocytes [12]. We conducted a randomized, double-blind, placebo-controlled study of Chlorella in middle-aged and older adults and found that eight weeks of Chlorella supplementation increased lutein levels and reduced phospholipid hydroperoxide (PLOOH) levels in erythrocyte membranes, indicating that Chlorella is useful for preventing senile dementia by maintaining normal erythrocyte function [13].

In Japan, ascidians (*Halocynthia roretzi*), a protochordate species, are mainly cultivated for food in the Miyagi and Hokkaido prefectures. Ascidians have particularly high levels of ethanolamine plasmalogen (PlsEtn) with docosahexaenoic acid (DHA) in their chemical structure, which is found in abundance in the human brain [7,14]. We previously reported in vitro and in vivo studies in which ascidian PlsEtn suppressed neuroblast apoptosis [15], inhibited amyloid-β aggregation [16], and improved cognitive function [17].

Brain-derived neurotrophic factor (BDNF) is a key molecule expressed in the central nervous system and intestines of humans and rodents and is involved in plastic changes related to learning and memory [18]. Changes in BDNF protein expression are associated with the onset of neurodegenerative diseases, such as dementia. BDNF promotes neuronal formation and growth mainly by binding to tropomyosin receptor kinase B (TrkB), a specific receptor on target cell surfaces, leading to TrkB autophosphorylation and activation of signaling molecules such as cAMP response element-binding protein (CREB) (BDNF–TrkB–CREB signaling pathway, Figure 1) [19]. Many studies on the beneficial effects of foods on memory impairment using animal models, such as AD models, have evaluated their association with BDNF signaling pathway activation, that is, BDNF protein expression, pTrkB/TrkB ratio, and pCREB/CREB ratio [20,21,22]. Since BDNF expression is decreased in the hippocampal and temporal lobes of patients with AD or AD model rats, measuring BDNF, TrkB, and CREB expression levels may be useful for assessing AD progression [23,24].

Glutamate receptors play an important role in the hippocampus. They are associated with functions such as memory, learning, signal transduction, neuronal plasticity, and protein synthesis [25]. Disruptions in these glutamate receptors have been observed in the hippocampus of AD patients, suggesting their role in cognitive dysfunction and memory loss due to the inefficient uptake and recycling system of glutamate in neurons [26]. In the hippocampus, glutamate receptors can be categorized into three subtypes: non-N-methyl-D-aspartate (NMDA) glutamate receptors (glutamate receptor 1 (GluR1), glutamate receptor 2 (GluR2), glutamate receptor 7 (GluR7)), NMDA glutamate receptors (NMDA glutamate receptor 1 (NR1), NMDA glutamate receptor 2B (NR2B)), and cAMP-type glutamate receptors (metabotropic glutamate receptor 3 (mGluR3)). Recent studies have reported interactions between these glutamate receptors and BDNF [27]. Activation of glutamate receptors promotes the release of BDNF from neurons. Conversely, BDNF has been shown to enhance glutamate receptor function by inducing phosphorylation of receptor subunits [28]. It is hypothesized that this interaction is disrupted in AD. Therefore, evaluating the expression levels of glutamate receptors may be potentially useful for assessing the progression of AD.

As described above, it is expected that the continuous combined consumption of lutein from Chlorella and PlsEtn from ascidian has the potential to be effective in AD. However, there are no studies evaluating the combined effects of these compounds, and the effects on the BDNF–TrkB–CREB signaling pathway and glutamate receptors remain unknown. Since food is not a single molecule but a complex of many molecules, elucidating this could provide important insights in this field. Therefore, this study investigated the synergistic effects of Chlorella and PlsEtn from ascidian on the activation of the hippocampal BDNF–TrkB–CREB signaling pathway and glutamate receptors in Sprague-Dawley rats.

## 2. Results

In the Con (control), CHL (*Chlorella pyrenoidosa* powder (200 mg/day/rat)), HRE (*Halocynthia roretzi* plasmalogen (0.07 mg/day/rat)), and Mix groups (CHL + HRE, *Chlorella pyrenoidosa* powder (200 mg/day/rat) + *Halocynthia roretzi* plasmalogen (0.07 mg/day/rat)), the hippocampus was excised from the brain, and BDNF signaling-related proteins expression and phosphorylation were detected by Western blotting. The hippocampal BDNF protein expression level in the Mix group was significantly higher than those in the CHL or HRE groups but showed an increasing trend compared to the Con group (Figure 2A). Moreover, the phosphorylation of TrkB, the specific receptor for BDNF, was significantly higher in the Mix group than in the Con and the CHL groups (Figure 2B). Further, phosphorylation of CREB was also significantly increased in the Mix group compared to the Con, CHL, and HRE groups (Figure 2C). These results indicate that the BDNF–TrkB–CREB signaling pathway was most activated in rats that received both *Chlorella pyrenoidosa* powder and *Halocynthia roretzi* extract.

The data of glutamate receptor expression are shown in Figure 3. No significant differences were observed between the Con, CHL, HRE, and Mix groups for any of the glutamate receptor subgroups (pGluR1 (A), pGluR2 (B), GluR7 (C), NR1 (D), pNR2B (E), and mGluR3 (F) in Figure 3) investigated in this study.

## 3. Discussion

The results of this study showed that the BDNF–TrkB–CREB signaling pathway in the hippocampus of Sprague-Dawley rats was most strongly activated when a combination of *Chlorella pyrenoidosa* powder and *Halocynthia roretzi* extract was administered. *Chlorella pyrenoidosa* contains high levels of lutein, a xanthophyll (200–300 mg per 100 g of dry weight). Lutein is a major antioxidant component in erythrocyte membranes. As described above, Chlorella intake increases lutein levels and decreases PLOOH in erythrocyte membranes in middle-aged and older adults [13]. Increased PLOOH levels in erythrocyte membranes reduce gas exchange with tissues through the membranes, indicating that decreased CO_2_ recovery and O_2_ supply in the brain are associated with cognitive decline [29]. Measurements of lutein levels in the brains of older adults aged 80 years or older revealed a significant correlation between lutein levels and antemortem cognitive function [30], suggesting that lutein intake is important for preventing cognitive decline associated with aging. Stringham et al. reported that lutein and zeaxanthin supplementation (13 mg/day) for 6 months increased serum BDNF levels [31]. Regarding this mechanism, decreased BDNF expression through oxidative and inflammatory stresses suggests the involvement of lutein in antioxidative and anti-inflammatory effects.

The lutein intake from the dose of Chlorella powder used in this study was 0.604 mg/day per rat. In addition to lutein, the carotenoids in the Chlorella powder include α-carotene and β-carotene (Table 1); however, lutein is likely involved in terms of its content. Although this study used Chlorella as a lutein-rich food, pure lutein may be similarly effective. Further studies are required to investigate the efficacy of pure lutein and other foods containing lutein. The mechanism by which plasmalogen activates the BDNF signaling pathway is unclear. In our study with AD model rats intracerebrally injected with amyloid-β, the oral administration of *Halocynthia roretzi* plasmalogen increased DHA-containing plasmalogen in the plasma, erythrocytes, liver, and cerebral cortex [17], suggesting that administered plasmalogen may reach the brain and function or increase the level of plasmalogen in the brain. Che et al. reported activation of the BDNF signaling pathway by plasmalogen administration with eicosapentaenoic acid (EPA) to AD model rats and scallop-derived plasmalogen administration to mice [32]. However, further studies are needed to elucidate the mechanism of activation of the BDNF signaling pathway by plasmalogen.

As shown in Table 1, the *Chlorella pyrenoidosa* powder used in this study contains not only lutein but also other various food components (protein, fat, ash, carbohydrate, dietary fiber, sodium, phosphorus, iron, calcium, potassium, magnesium, zinc, vitamin B1, vitamin B2, vitamin B6, vitamin B12, vitamin D2, vitamin E, folate, biotin, inositol, α-carotene, β-carotene, lutein). Explaining the functions and mechanisms of “foods” with various food components is difficult to specify and requires significant cost and time due to its complexity. Furthermore, current research in food science and nutrition often takes an approach that focuses on only a single food component, even though it interacts with other multiple food components to influence physiological responses [33]. On the other hand, with the recent development of AI (artificial intelligence) technology, new approaches are being developed that can evaluate and explain the health functions of such “foods” composed of multiple molecules [33]. Zhao et al. developed an AI model for finding food molecules that exert anticancer effects based on similarities between anticancer drugs and food ingredients and proposed an approach that incorporates seven types of machine learning with a soft voting algorithm [34]. By using this AI model, the accuracy of the dataset of 50 food components considered was improved from 82% to 87%, resulting in improved accuracy in predicting their anticancer effects. Kozawa et al. developed an AI model that can predict the adverse effects of drugs and indications for treatment by combining experimental approaches (multi-organ transcriptome data from mice and human clinical outcome datasets) with machine learning approaches [35]. Guasch-Ferre et al. analyzed 385 walnut-derived metabolites in human plasma using liquid chromatography coupled with tandem mass spectrometry and tried to identify the relationship between these metabolites and type 2 diabetes and cardiovascular disease risk using machine learning approaches [36]. As a result, this AI model showed that 19 metabolites associated with walnut consumption were related to a reduced risk of type 2 diabetes and cardiovascular disease. These advancements in AI technology for food science highlight the potential of machine learning and AI in understanding food components and their health benefits. There is potential in applying these AI models to the effects of simultaneous intake of Chlorella and ascidian plasmalogen used in this study on the BDNF–TrkB–CREB signaling pathway. Furthermore, AI techniques might be utilized to explain how the combination of Chlorella and ascidian plasmalogen interacts and contributes to overall physiological effects and to predict potential synergistic effects.

In this study, the BDNF–TrkB–CREB signaling pathway was activated by combined treatment at doses equal to those that were less effective as monotherapies, despite a relatively short treatment period of one week. The mechanism underlying the combined effects of Chlorella lutein and ascidian plasmalogens remains unknown. Kotake-Nara et al. reported that intestinal absorption of carotenoids by Caco-2 cells, although low in general, was enhanced by glycerophospholipids in mixed micelles [37], suggesting that phospholipids containing plasmalogen promote lutein absorption. In addition, it has been reported that vitamin B12 promotes plasmalogen biosynthesis and protects plasmalogen from oxidative stress in neuroblastoma cells [38]. Chlorella powder used in this study contained 200 µg/100 g of vitamin B12 (Table 1), and it is estimated that rats in the CHL and Mix groups consumed approximately 40% more vitamin B12 compared to rats in other groups. Therefore, vitamin B12 in Chlorella may have contributed to the plasmalogen biosynthesis and protection of plasmalogen from oxidative stress in the rat brain tissue.

Studies show that the decrease in PlsEtn in the brains of patients with AD varies among the molecular species; PlsEtn bearing DHA or arachidonic acid, which are highly unsaturated fatty acids, decreases significantly [39]. One study reported that PlsEtn bearing DHA has a powerful anti-apoptotic effect on neuroblasts compared with other plasmalogen species. In this mechanism, excessive oxidative and inflammatory stresses activate phospholipase A_2_ (PLA_2_), including PlsEtn-selective PLA_2_ in nerve cells, which promotes the cleavage of DHA from PlsEtn; the resulting DHA inhibits the activation of cPLA_2_ and COX-2, as well as inhibiting apoptosis caused by the arachidonic acid cascade [15]. PlsEth has also been reported to inhibit amyloid-β aggregation and promote degradation [16]. As already mentioned above, the effect of plasmalogen on BDNF expression has been reported by Che et al. [32]. Plasmalogen with EPA activates the BDNF–TrkB–CREB signaling pathway in AD model rats treated with amyloid-β. However, the mechanism underlying its activation has not been elucidated. In this study, consumption of *Chlorella pyrenoidosa* powder and *Halocynthia roretzi* extract did not result in significant differences in the expression of glutamate receptors in the hippocampus of Sprague-Dawley rats.

However, it is the phosphorylation of GluR1,2 that shows the potential for an upward trend in the same direction as the activation of the BDNF system. This is the AMPA receptor (α-amino-3-hydroxy-5-methyl-4-isoxazolepropionic acid receptor), an ion channel-type glutamate receptor that is predominantly present at excitatory synapses in the central nervous system. Activation of AMPA receptors causes excitatory synaptic transmission. Excitatory synaptic transmission is important for information processing, learning, and memory formation. In other words, it is expected that activation of the BDNF system may lead in the direction of an upward trend in GluR1 and GluR 2, and these may lead to information processing, learning, and memory formation. On the other hand, it is conceivable that NR1 and NR2B of the NMDA receptor may go in the opposite direction of the downward trend as GluR1 and GluR2. Also, it could be suggested to study the activation of kinases involved in the phosphorylation of glutamate receptor subunits [40]. In the C-terminal region of AMPA and NMDA receptor subunits, serine, threonine, and tyrosine residues have been identified as the main sites susceptible to phosphorylation. These sites are known to undergo activity-dependent or constitutive phosphorylation induced by changes in cellular and synaptic signals [40]. Based on the results of this study, it is necessary to resolve the above-mentioned expectations in the future. Another consideration is that there is a possibility of different effects and variations, particularly in the expression and phosphorylation of glutamate receptors, between AD model rats and Sprague-Dawley rats. This might be due to the fact that AD model rats are designed to mimic the pathophysiology of AD with specific genetic mutations and pathway alterations. Therefore, further investigation, including the use of AD model rats, will be necessary to confirm the effects on these receptors.

## 4. Materials and Methods

### 4.1. Materials

We used ascidian PlsEtn extract (Yaizu Suisankagaku Industry Co., Ltd., Shizuoka, Japan). Ascidian extract was prepared as follows: crude lipid (total lipid fraction) was extracted from dried ascidian (*Halocynthia Roretzi*) with acetone or hexane, and PlsEtn extract (phospholipid fraction) was obtained by purification with hexane-ethanol. The ascidian extract contained 98% total lipid, 39% phospholipids, 5% PlsEtn, and a small amount of choline plasmalogen (Table 2). Dried *Chlorella pyrenoidosa* powder was provided by Sun Chlorella Corp. (Kyoto, Japan). The nutritional composition of dried Chlorella powder is shown in Table 1.

### 4.2. Animals and Treatment

Male Sprague-Dawley rats (six weeks old) were purchased from SLC Inc. (Shizuoka, Japan) and individually housed in standard polycarbonate cages for 7 days. A normal diet (MF; Oriental Yeast Manufacturing Co., Ltd., Tokyo, Japan) and water were available ad libitum during acclimatization. Rats were kept in a room maintained at 23 ± 2 °C. After acclimatization for one week, the rats were assigned to one of four groups (five rats per group): control (Con; soybean oil/canola oil + MF diet), Chlorella (CHL; soybean oil/canola oil + diet containing Chlorella), ascidian extract (HRE; ascidian extract + MF diet), or Chlorella + ascidian extract (Mix; ascidian extract + diet containing Chlorella).

The ascidian extract solution was prepared by diluting the ascidian extract with soybean and canola oils (Nisshin OilliO Group, Ltd., Tokyo, Japan) to a 0.07 mg/day dose of ethanolamine plasmalogen (administration solution volume 0.5 mL) and was administered into the stomach once daily for one week via gavage feeding tube. MF diet containing 1% Chlorella powder was administered at 200 mg/day for one week.

This study was conducted in accordance with the “Guide for the Care and Use of Laboratory Animals” (NIH Publication No. 85-23, revised in 1996). All experimental protocols were approved by the Ethics Committee on Animal Use of Suzuka University of Medical Science (Approval No.56, Approval Date: 7 June 2016).

### 4.3. Western Blotting

The rats were euthanized on the day after the final administration, and their brains were rapidly excised. Hippocampal tissues were stored at −70 °C until Western blotting was performed to examine BDNF signaling-related protein expression and phosphorylation.

Western blot analysis of hippocampal protein expression and phosphorylation was performed as described previously [41]. Homogenized hippocampal lysates were separated using 10% sodium dodecyl sulfate-polyacrylamide gel electrophoresis (SDS-PAGE) and transferred to Amersham Hybond P PVDF 0.45 membranes (Bio-Rad, Hercules, CA, USA; Cytiva, Tokyo, Japan). Membranes were incubated with 5% bovine serum albumin (BSA) in TBST for 1 h, followed by incubation with primary antibodies against BDNF (ab108319), TrkB (#4603), pTrkB (#4619), pCREB (06-519), CREB (#9197), pGluR1 (04-823), pGluR2 (ab52180), GluR7 (ab183035), mGluR3 (ab140741), pNR2B (AB5403), and β-actin (#4970) at a 1:1000 dilution overnight at 4 °C. Anti-BDNF antibody was purchased from Abcam (Cambridge, UK) and Merck (Darmstadt, Germany) and other primary antibodies were purchased from Cell Signaling Technology (Tokyo, Japan). This was followed by incubation with horseradish peroxidase-conjugated secondary antibodies (7074, 1:1000, Cell Signaling Technology) for 1 h. Immunoreactive bands were detected using an enhanced chemiluminescence detection kit, and a Light Capture AE-6971/2 device (ATTO Corp., Tokyo, Japan) was used for visualization. Band intensities were normalized to β-actin, total TrkB, CREB, GluR1 (04-855), GluR2 (#5306), or NR2B (#4207) using a CS Analyzer 4 (ATTO Corp, Tokyo, Japan).

### 4.4. Statistical Analysis

Values are expressed as means ± standard error (SE) and were derived from measurements of five rats. All statistical analyses were performed using SPSS version 25 (IBM Corp, Armonk, NY, USA). The homogeneity of variance was checked using Levene’s test. One-way ANOVA was used for intergroup comparisons, and when ANOVA revealed significant differences, Dunnett’s t- or T3 post hoc tests were used to identify significant differences between the two groups. Statistical significance was set at *p* < 0.05.

## 5. Conclusions

In summary, we examined the effects of Chlorella and PlsEtn from ascidian, which are anticipated to be effective in preventing dementia, both alone and in combination, on the activation of the BDNF–TrkB–CREB signaling pathway, which is known to be decreased in AD patients.

The present study provides the first evidence that the combination of Chlorella and ascidian PlsEtn enhanced activation of the BDNF–TrkB–CREB signaling pathway. The BDNF–TrkB–CREB signaling pathway was activated by their combination at doses that were not as effective as monotherapy, despite a relatively short treatment period of one week. These results suggest that the combination of Chlorella and ascidian PlsEtn is expected to improve cognitive function through different mechanisms and may have a synergistic preventive effect in dementia patients. We plan to examine this effect in future studies, such as oral intake studies in humans.

## Figures and Tables

**Figure 1 molecules-29-00357-f001:**
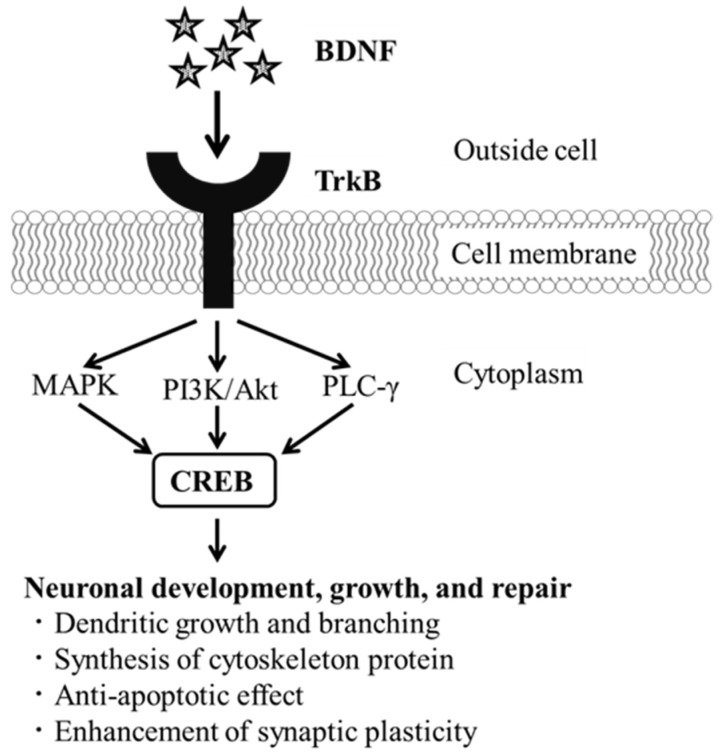
Signaling pathways mediated by BDNF in neuronal development, growth, and repair. BDNF, brain-derived neurotrophic factor; CREB, cAMP response element-binding protein; MAPK, mitogen-activated protein kinase; PI3K/Akt, phosphatidylinositol-3 kinase/protein kinase B; PLC-γ, phospholipase Cγ; TrkB, tropomyosin receptor kinase B. This figure has been modified with permission from Ref. [19] (Palasz et al. 2020) under the Creative Commons CC BY 4.0 license.

**Figure 2 molecules-29-00357-f002:**
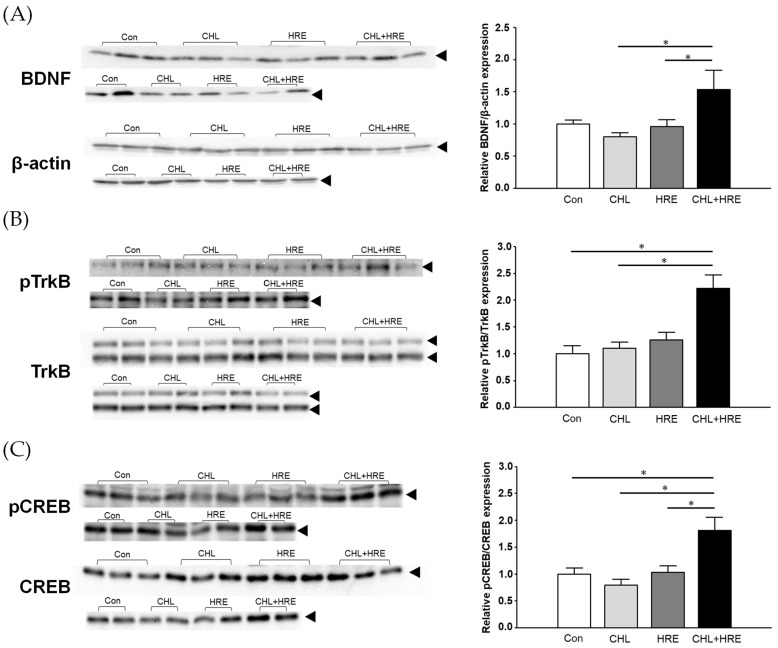
Effects of *Chlorella pyrenoidosa* powder, *Halocynthia roretzi* extract, and a mixture of both on hippocampal brain-derived neurotrophic factor (BDNF) signaling. Hippocampal protein (**A**) BDNF, (**B**) phospho-tropomyosin receptor kinase B (pTrkB), and (**C**) phospho-cAMP response element-binding protein (pCREB) levels were determined by Western blotting. BDNF expression was normalized to β-actin expression, whereas pTrkB and pCREB levels were normalized to those of TrkB and CREB, respectively. Each value is presented as a ratio relative to that of the control group (meaning that the ratio is arbitrarily considered as 1 for the control group). Data are presented as means ± SE; *n* = 5; the data come from two experimental series (3 + 2); for TrkB, the intensity of the two bands added before the ratio; * *p* < 0.05. Con, control; CHL, *Chlorella pyrenoidosa* powder (200 mg/day/rat); HRE, *Halocynthia roretzi* plasmalogen (0.07 mg/day/rat); CHL + HRE, *Chlorella pyrenoidosa* powder (200 mg/day/rat) + *Halocynthia roretzi* plasmalogen (0.07 mg/day/rat).

**Figure 3 molecules-29-00357-f003:**
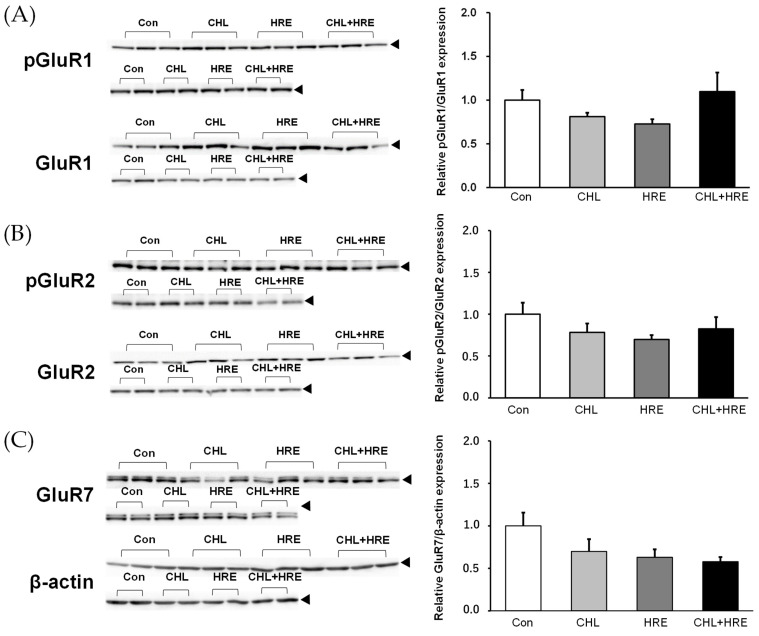
Effects of *Chlorella pyrenoidosa* powder, *Halocynthia roretzi* extract, and a mixture of both on hippocampal glutamate receptors. Hippocampal (**A**) pGluR1, (**B**) pGluR2, (**C**) GluR7, (**D**) NR1, (**E**) pNR2B, and (**F**) mGluR3 levels were determined by Western blotting. GluR7, NR1, and mGluR2 expressions were normalized to β-actin expression, whereas pGluR1, pGluR2, and pNR2B levels were normalized to those of GluR1, GluR2, and NR2B, respectively. Each value is presented as a ratio relative to that of the control group (meaning that the ratio is arbitrarily considered as 1 for the control group). Data are presented as *n* = 5. The data come from two experimental series (3 + 2). Con, control; CHL, *Chlorella pyrenoidosa* powder (200 mg/day/rat); HRE, *Halocynthia roretzi* plasmalogen (0.07 mg/day/rat); CHL + HRE, *Chlorella pyrenoidosa* powder (200 mg/day/rat) + *Halocynthia roretzi* plasmalogen (0.07 mg/day/rat). GluR1, glutamate receptor 1; GluR2, glutamate receptor 2; GluR7, glutamate receptor 7; mGluR3, metabotropic glutamate receptor 3; NR1, cell signaling NMDA receptor 1; NR2B, NMDA receptor 2B; pGluR1, phosphorylated glutamate receptor 1; pGluR2, phosphorylated glutamate receptor 2; pNR2B, phosphorylated NMDA receptor 2B.

**Table 1 molecules-29-00357-t001:** Nutrient composition of dried *Chlorella pyrenoidosa* powder.

Composition	(g/100 g Powder)
Protein	59.4 g
Fat	10.8 g
Ash	9.2 g
Carbohydrate	3.6 g
Dietary fiber	13.5 g
Sodium	55.0 mg
Phosphorus	1.97 g
Iron	180.0 mg
Calcium	1.26 g
Potassium	1.02 g
Magnesium	417 mg
Zinc	1.11 mg
Vitamin B1	1.50 mg
Vitamin B2	5.06 mg
Vitamin B6	1.74 mg
Vitamin B12	0.20 mg
Vitamin D2	1.46 mg
Vitamin E	7.10 mg
Folate	1.50 mg
Biotin	0.229 mg
Inositol	137 mg
α-carotene	23.5 mg
β-carotene	56.4 mg
Lutein	302 mg

**Table 2 molecules-29-00357-t002:** Total lipids and PlsEtn in ascidian extract.

Composition	(g/100 g Extract)
Total lipids	98
Phospholipids	39
PlsEtn	
18:0/18:1-PlsEtn	0.150
18:0/20:4-PlsEtn	0.523
18:0/20:5-PlsEtn	3.126
18:0/22:6-PlsEtn	1.742

PlsEtn: ethanolamine plasmalogen.

## Data Availability

The data that support the findings of this study are available from the corresponding authors, upon reasonable request.

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
