# Peer review of "Simultaneous Intake of Chlorella and Ascidian Ethanolamine Plasmalogen Accelerates Activation of BDNF–TrkB–CREB Signaling in Rats"

_molecules, 2024, doi:10.3390/molecules29020357_

Round 1

Reviewer 1 Report

Comments and Suggestions for Authors

Comments

The paper entitled “Simultaneous intake of Chlorella and ascidian ethanolamine plasmalogen accelerates activation of hippocampal neurotrophin BDNF-TrkB-CREB signaling in SD rats” written by Hideo Takekoshi investigated the effects of combining lutein and PlsEtn using lutein-rich Chlorella and ascidian extracts containing high levels of PlsEtn bearing docosahexaenoic acid on the activation of the BDNF-TrkB-CREB signaling pathway in the hippocampus of Sprague-Dawley rats. The results of this study suggest that the combination of Chlorella and ascidian PlsEtn may have a preventive effect against dementia. Generally, this article is well prepared and meets the requirement of this journal.

Since consumption of Chlorella pyrenoidosa powder and Halocynthia roretzi extract did not result in significant differences in the expression of glutamate receptors in the hippocampus of Sprague-Dawley rats, rational discussions are strongly suggested to provided based on the literatures as well as experimental data. Above all, I recommend this paper to be published after minor revision.

Some points:

1.     Figure 2 (especially for western blotting) need be improved and those with high quality should be provided.

2.     Line 113, that---those

3.     Lines 146-147, pGluR1, pGluR2 and pNR2B---pGluR1, pGluR2, and pNR2B

GluR1, GluR2 and NR2B---GluR1, GluR2, and NR2B

4.     Line 203, Total lipids, and--- Total lipids and

5.     The format of Table 2 should be improved.

6.     Please check the styles of all the references. For example, Page of ref29 is absent.

Author Response

Comments from Reviewer 1

The paper entitled “Simultaneous intake of Chlorella and ascidian ethanolamine plasmalogen accelerates activation of hippocampal neurotrophin BDNF-TrkB-CREB signaling in SD rats” written by Hideo Takekoshi investigated the effects of combining lutein and PlsEtn using lutein-rich Chlorella and ascidian extracts containing high levels of PlsEtn bearing docosahexaenoic acid on the activation of the BDNF-TrkB-CREB signaling pathway in the hippocampus of Sprague-Dawley rats. The results of this study suggest that the combination of Chlorella and ascidian PlsEtn may have a preventive effect against dementia. Generally, this article is well prepared and meets the requirement of this journal.

              Since consumption of Chlorella pyrenoidosa powder and Halocynthia roretzi extract did not result in significant differences in the expression of glutamate receptors in the hippocampus of Sprague-Dawley rats, rational discussions are strongly suggested to provided based on the literatures as well as experimental data. Above all, I recommend this paper to be published after minor revision.

Answer to Reviewer 1:

We highly appreciate your accurate advice for preparation of this article to be better than before. We have revised the points you pointed out in this revised manuscript by using Microsoft Word's revision history function. We would like you to confirm the revised comment and manuscript below.

Comment 1

Figure 2 (especially for western blotting) need be improved and those with high quality should be provided.

Answer:

Thanks for your comment. We have replaced Figure2 with high quality figure. In addition, we have replaced Figure3 with high quality one as well. (Figure 2 and Figure 3 in revised manuscript)

Comment 2

Line 113, that---those

Answer:

Thanks for your comment. We have revised it. (Line 120 in revised manuscript)

Comment 3

Lines 146-147, pGluR1, pGluR2 and pNR2B---pGluR1, pGluR2, and pNR2B

GluR1, GluR2 and NR2B---GluR1, GluR2, and NR2B

Answer:

Thanks for your comment. We have revised it. (Line 164-165 in revised manuscript)

Comment 4

Line 203, Total lipids, and--- Total lipids and

Answer:

Thanks for your comment. We have revised it. (Line 298 in revised manuscript)

Comment 5

The format of Table 2 should be improved.

Answer:

Thanks for your comment. As reviewer 1 said, the format and quality of Table 2 was poor. Therefore, we replaced it with a table with improved format and new information. (Table 2 in revised manuscript)

Comment 6

Please check the styles of all the references. For example, Page of ref29 is absent.

Answer:

Thanks for your comment. We confirmed the style, lack of page number of references and revised it. (Reference 15, 23, and 41 in revised manuscript)

Reviewer 2 Report

Comments and Suggestions for Authors

This manuscript describes the combined effects of Chlorella and ascidian extracts, respectively rich in lutein and PlsEtn plasmalogen. This work complements previous papers from the same research team on the effects of plasmalogen on various pathologies compiled by the same group (ref 6 and 12-16). It describes that the plasmalogen effect could be enhanced by the co-administration of Chlorella extract, at least on the hippocampus BDNF concentration. Unfortunately, the assumption that it could also influence the status of glutamate receptors could not be demonstrated.

I have some comments about this submission.

Introduction:

  1. Line 87: A more recent publication could be cited [e.g., Kim J. et al, Int J Mol Sci. 2019 Jun; 20(12): 2943. doi: 10.3390/ijms20122943].
  2. Reference needed for line 90: "… inducing phosphorylation of receptor subunits (?)."
  3. Lines 93-94: "As described above, continuous consumption of lutein from Chlorella and PlsEtn from ascidian has the potential to be effective in AD." It is not clear whether the effect of this co-consumption was demonstrated or expected.

Results:

  1. In Figure 1: If possible, the resolution of the western blot must be increased, and the molecular weight of each protein studied indicated. I suppose that the data come from two experimental series (3 + 2); if yes, this must be indicated.
  2. For TrkB, are the intensity of the 2 bands indicated added before the ratio?
  3. "Ratio relative to that of the control group" means that the ratio is arbitrarily considered as 1 for the control group?
  4. In Figure 3: In the legend, BDNF expression is mentioned but not represented in the figure (line 145). Same comment as for Figure 2 about molecular weight indication.

Discussion:

In my view, the discussion must be more focused on the results gained by this study, although there are controversies. The increase in the activation of the BDNF signaling pathway is not correlated with the expected glutamate receptor expression or activation. I am sure the authors have many suggestions to explore this point; it would be interesting if they shared their hypotheses with the readers.

As an example, it could be suggested to study the activation of kinases involved in the phosphorylation of glutamate receptor subunits (see, for example, Wang et al; Eur J Pharmacol. 2014 Apr 5; 728: 183–187. doi: 10.1016/j.ejphar.2013.11.019).

Author Response

Comments from Reviewer 2

This manuscript describes the combined effects of Chlorella and ascidian extracts, respectively rich in lutein and PlsEtn plasmalogen. This work complements previous papers from the same research team on the effects of plasmalogen on various pathologies compiled by the same group (ref 6 and 12-16). It describes that the plasmalogen effect could be enhanced by the co-administration of Chlorella extract, at least on the hippocampus BDNF concentration. Unfortunately, the assumption that it could also influence the status of glutamate receptors could not be demonstrated.

I have some comments about this submission.

Dear Reviewer 2:

Thank you very much for reviewing this manuscript. All the comments from reviewer 2 are very important and essential for this manuscript. Thank you again for spending your valuable time on this manuscript. We made changes to this manuscript according to suggestions from reviewer 2. We have revised the points you pointed out in this revised manuscript by using Microsoft Word's revision history function. We would like you to confirm the revised comment and manuscript below.

Introduction: Comment 1

Line 87: A more recent publication could be cited [e.g., Kim J. et al, Int J Mol Sci. 2019 Jun; 20(12): 2943. doi: 10.3390/ijms20122943].

Answer:

Thank you for providing important publication for this article. We have cited the publication suggested by reviewer 2 as a new reference [27]. (Line 93 in revised manuscript)

Introduction: Comment 2

Reference needed for line 90: "… inducing phosphorylation of receptor subunits (?)."

Answer:

Thank you for pointing this out. As referee 2 said, reference was needed. We have cited the reference [28] there. (Line 96 in revised manuscript)

Introduction: Comment 3

Lines 93-94: "As described above, continuous consumption of lutein from Chlorella and PlsEtn from ascidian has the potential to be effective in AD." It is not clear whether the effect of this co-consumption was demonstrated or expected.

Answer:

Thank you for pointing this out. As reviewer 2 said, this part of the description was ambiguous and misleading to the reader. Since the effect had not yet been demonstrated, we used the word "expected". (Line 99-100 in revised manuscript)

Results: Comment 1

In Figure 1: If possible, the resolution of the western blot must be increased, and the molecular weight of each protein studied indicated. I suppose that the data come from two experimental series (3 + 2); if yes, this must be indicated.

Answer:

Thank you for your comment. As reviewer 2 mentioned, it comes from two experimental series. This explanation was missing. So, we have added it. (Line 144 and 159 in revised manuscript)

Results: Comment 2

For TrkB, are the intensity of the 2 bands indicated added before the ratio?

Answer:

Thank you for your comment. TrkB uses the value calculated by adding the two bands together before showing the intensity ratio. As reviewer 2 pointed out, this explanation was missing, so we have added it. (Line 144 in revised manuscript)

Results: Comment 3

"Ratio relative to that of the control group" means that the ratio is arbitrarily considered as 1 for the control group?

Answer:

Thank you for your comment. As Reviewer 2 pointed out, meaning that the ratio of the control group is arbitrarily considered to be 1. This explanation was missing and we have added it. (Line 143 and 158-159 in revised manuscript)

Results: Comment 4

In Figure 3: In the legend, BDNF expression is mentioned but not represented in the figure (line 145). Same comment as for Figure 2 about molecular weight indication.

Answer:

Thank you for your comment. There is no data on BDNF expression in Figure 3, which was inappropriate. The data should be GluR7, NR1, and mGluR2. We have revised this legend. (Line 155-156 in revised manuscript)

Discussion:

In my view, the discussion must be more focused on the results gained by this study, although there are controversies. The increase in the activation of the BDNF signaling pathway is not correlated with the expected glutamate receptor expression or activation. I am sure the authors have many suggestions to explore this point; it would be interesting if they shared their hypotheses with the readers.

As an example, it could be suggested to study the activation of kinases involved in the phosphorylation of glutamate receptor subunits (see, for example, Wang et al; Eur J Pharmacol. 2014 Apr 5; 728: 183–187. doi: 10.1016/j.ejphar.2013.11.019).

Answer:

We also felt that the discussion in this paper was cheap and needed more improvement. We thank reviewer 2 for pointing this out. We have added our discussion and also added a discussion of the importance of studying the activation of kinases involved in the phosphorylation of glutamate receptor subunits, which reviewer 2 commented (Line 190-249, 265-282 in revised manuscript). Also, we added We have cited the publication suggested by reviewer 2 as a new reference [40]. (Line 280 in revised manuscript)

Reviewer 3 Report

Comments and Suggestions for Authors

Dear Authors,

Your Article entitled "Simultaneous intake of Chlorella and ascidian ethanolamine plasmalogen accelerates activation of hippocampal neurotrophin BDNF-TrkB-CREB signaling in SD rats" has been Reviewed,

This Article deserves attention since it highlights a very important topic related to the Human Health,

Kindly find below a list of my Remarks and Comments concerning your work:

01- Concerning the Title of the Article, Authors are invited to change this title since it is too long and it contains a lot of abbreviations.

02- Concerning the References in the text, when authors used numerous consecutive references for example Reference 3, 4, 5 and 6, They are invited to put it as follow: [3-6] instead of [3, 4, 5, 6]

03- Concerning the Figure 2, Authors are invited to put all sub-figures (A, B and C) in addition to the figure's legends on the same page.

04- Concerning the Figure 3, Authors are invited to put all sub-figures (A, B, C, D, E and F) in addition to figure's legends on the same page.

05- Concerning the Table 2, Authors are invited to remove the empty row between "Dietary fiber" and "Lutein". They are also invited to align the numbers.

06- In the Whole manuscript, Authors are invited to replace "p" value by "P" value.

07- In the Figures, some legends are of different font and of different font size. Authors are invited to make it homogenous.

08- In some figures, for example in the Figure 2C, the Western is not very clear, Authors are invited to put figures with better resolution. 

Best Regards,

Author Response

Comments from Reviewer 3

Your Article entitled "Simultaneous intake of Chlorella and ascidian ethanolamine plasmalogen accelerates activation of hippocampal neurotrophin BDNF-TrkB-CREB signaling in SD rats" has been Reviewed,

This Article deserves attention since it highlights a very important topic related to the Human Health,

Kindly find below a list of my Remarks and Comments concerning your work:

Dear Reviewer 3:

Thank you very much for reviewing this article. All the comments from reviewer 3 are very important and essential for this article. Thank you again for spending your valuable time on this article. We made changes to this article according to suggestions from reviewer 3. We have revised the points you pointed out in this revised manuscript by using Microsoft Word's revision history function. We would like you to confirm the comment and manuscript below.

Comment 1

Concerning the Title of the Article, Authors are invited to change this title since it is too long and it contains a lot of abbreviations.

Answer:

Thanks for your comment. As reviewer 3 said, we thought the title had too many abbreviations and was too long. Therefore, re revised the title “Simultaneous intake of Chlorella and ascidian ethanolamine plasmalogen accelerates activation of hippocampal neurotrophin BDNF-TrkB-CREB signaling in SD rats” to “Simultaneous intake of Chlorella and ascidian ethanolamine plasmalogen accelerates activation of BDNF-TrkB-CREB signaling in rats”. (Line 2-4 in revised manuscript)

Comment 2

Concerning the References in the text, when authors used numerous consecutive references for example Reference 3, 4, 5 and 6, They are invited to put it as follow: [3-6] instead of [3, 4, 5, 6]

Answer:

Thanks for your comment. We revised the numerous consecutive references into [number-number]. (Line 50 and 79 in revised manuscript)

Comment 3

Concerning the Figure 2, Authors are invited to put all sub-figures (A, B and C) in addition to the figure's legends on the same page.

Answer:

Thanks for your comment and we agree with it. We have replaced Figure2 with high quality figure. Each sub-figures and figure's legends were put in same page. (Line 133-148 in revised manuscript)

Comment 4

Concerning the Figure 3, Authors are invited to put all sub-figures (A, B, C, D, E and F) in addition to figure's legends on the same page.

Answer:

Thanks for your comment and we agree with it. We have replaced Figure3 with high quality figure. Each sub-figures and figure's legends were put in same page. (Line 149-166 in revised manuscript)

Comment 5

Concerning the Table 2, Authors are invited to remove the empty row between "Dietary fiber" and "Lutein". They are also invited to align the numbers.

Answer:

Thanks for your comment. As reviewer 3 said, the format and quality of Table 2 was poor. Therefore, we replaced it with a table with improved format and new information. (Table 2 in revised manuscript)

Comment 6

In the Whole manuscript, Authors are invited to replace "p" value by "P" value.

Answer:

Thanks for your comment. We revised “p” to “P”. (Line 145 and 347 in revised manuscript)

Comment 7

In the Figures, some legends are of different font and of different font size. Authors are invited to make it homogenous.

Answer:

Thanks for your comment. We have replaced each figure with a more unified one. (Each figures in revised manuscript)

Comment 8

In some figures, for example in the Figure 2C, the Western is not very clear, Authors are invited to put figures with better resolution.

Answer:

Thanks for your comment. We have replaced each figure with more high quality. (Each figures in revised manuscript)
